# Inter-Organisational Exercises in Dry and Wet Context—Why Do Maritime Response Organisations Gain More Knowledge from Exercises at Sea Than Those on Shore?

**Eric Carlström** [1,2,*], **Leif Inge Magnussen** [2], **Elsa Kristiansen** [2], **Johan Berlin** [3] **and Jarle Løwe Sørensen** [2]

[1] Institute of Health and Care Sciences, Sahlgrenska Academy, University of Gothenburg, SE-405 30 Goteborg, Sweden
[2] USN School of Business, University of South-Eastern Norway, P.O. Box 235, 3603 Kongsberg, Norway; leif.magnussen@usn.no (L.I.M.); elsa.kristiansen@usn.no (E.K.); jarle.sorensen@usn.no (J.L.S.)
[3] Department of Social and Behavioural Studies, University West, SE-461 86 Trollhättan, Sweden; johan.berlin@hv.se
* Correspondence: eric.carlstrom@gu.se; Tel.: +46-7027-38126

**Abstract:** This is a study of inter-organisational exercises arranged by on-shore organisations (ONSOs) and off-shore organisations (OFFSOs). The aim was to compare findings from trained emergency staffs' perceptions of the impact of exercises. The data were retrieved from surveys conducted by the research team in conjunction with exercises. The surveys included staff from the coast guard, sea rescue, police department, fire department and ambulance services. A total of 94 professional emergency personnel participated in the ONSO exercises and 252 in the OFFSO exercises. The study was based on the suggestion that collaborative elements during an inter-organisational exercise promote learning, and learning is important to make the exercises useful. Collaboration proved to be a predictor for some of the items in learning, and learning was a predictor for some of the items in utility. There was, however, a stronger covariation between collaboration, learning and utility in the OFFSOs exercises than in the ONSOs. One reason might be the different cultures of emergency staff involved in on-shore and off-shore organisations. The OFFSOs' qualifications may be dominated by seamanship, together with professional practice, and all parties are expected to act as first responders. ONSOs, on the other hand, practice exercises from a strict professional and legal perspective.

**Keywords:** exercises; learning; inter-organisational; off-shore; on-shore; emergencies

## 1. Introduction

Strategic sustainability infers a built-in resistance to the prominence and effects of crisis events. A common way to maximise society sustainability is by maintaining the emergency response via regular exercises. Inter-organisational exercises are supposed to help authorities to become better at handling accidents, crises and disasters. Exercises involving different emergency services are carried out with the purpose of strengthening the inter-organisational ability to deal with difficult events that require extensive resources in a short time. There are, however, few studies of the efficiency of inter-organisational exercises in terms of learning and utility. One exception is Scandinavia, where inter-organisational exercises at sea and ashore have been studied [1–5]. The concept of inter-organisational exercises, as it is used in Scandinavia, describes exercises aiming to prevent organisational fragmentation, and develop integration and distribution of tasks [6,7].

In this article, we compare data from nine exercises (three ONSOs and six OFFSOs) from five published studies in order to find similarities and differences in terms of learning and utility. Comparing inter-organisational exercises arranged by on-shore and off-shore organisations can reveal context specific challenges and differences in learning outcomes. When extrapolated, it can pin-point strengths and weaknesses in different contexts of emergency preparedness. Such knowledge can, in turn, offer suggestions on how to improve the outcome from exercises and emergency response.

During the management of a societal crisis, the need for collaboration between emergency services has been actualised. As a result of criticism towards the management of rescue work during the 2011 attacks on the government blocks in Oslo and the Labour Parties Youth League summer camp on the island of Utoya, Norway in 2012, an extra principle was added to the national emergency preparedness legislation. It was the principle of collaboration, which was supposed to facilitate inter-organisational actions during emergencies [8]. In Sweden, as well as in Norway, where the data for this study were collected, the governments emphasise the importance of exercises to develop collaboration on different societal levels. In particular, inter-organisational collaboration is highlighted as a particularly important task to be practiced, specifically by getting employees to take the initiative to help each other across organisational boundaries [9–12]. This is especially true during time constraining emergencies [13,14].

In several studies, the inclusion of collaborative elements in exercises has proved to contribute to learning, and learning contributes to usefulness in real life situations. However, the studies show that even if there was a significant learning effect because of collaboration, the impact was, in some cases, moderate [13,15,16] and emergency staff have difficulties in learning from past mistakes [17]. Even if emergency response organisations are supposed to collaborate, they tend to prioritise the specific tasks they are trained for, instead of seeing the big picture [18]. Staff may engage in tasks they are accustomed to but are inactive when unaccustomed tasks need to be performed [19]. This is contradictory to collaboration, i.e., only focusing on one's own responsibilities, but being prepared to take initiatives beyond [20].

Here, learning is studied at the collective level [21]. Different patterns of action are built within the organisations, where individuals share experiences, learn from each other, and develop common approaches [22]. Learning occurs from communication across borders, different organisational agendas, the use of common resources and the different skills of collaborating professionals. According to Stein, the knowledge can be shared by permeable boundaries, ensuring multifunctional networks and integration. Demarcated boundaries and hierarchical grouping where activities are distributed into segmented tasks, on the other hand, may prevent knowledge sharing [21]. Institutional learning is accomplished by repeated action patterns, which over time create institutions that are stable in nature [23]. In this study, learning is considered to be the effect of a successful inter-organisational exercise, and learning is assumed to have impact on actual practice. Successful exercises, in terms of learning, may integrate professions at the fictive accident site [24]. Therefore, learning may occur when new patterns of thought are constructed and considered useful [25,26]. During exercises, learning is assumed to stand for change and development [27] and contribute to being open to different options and encourages initiatives to collaborate with others to achieve improved results [28]. According to Stein, inter-organisational learning builds trust, team building and coherence crossing professional borders [21]. Hence, learning can facilitate the development of routines, rules and models that have influence on daily work [29,30]

## 2. Maritime and Land-Based Collaboration

The collaborative culture seems to be different between emergency staff in maritime versus land-based contexts. Collaboration in the maritime context crosses national and cultural borders and is, in contrast to land-based, regulated by bi- and multi-lateral agreements and treaties. One such example of collaboration is in the high north area, where resources for search and rescue missions are lacking. Even if some areas, such as coastal Norway close to the mainland, are well-developed, distant areas are wastelands that are occasionally trafficked by crowded cruise vessels and oil installations.

The risks related to going aground or collisions with ice, or fire are not in accordance with the number of available rescue services, especially in remote areas. Disasters in wastelands and distant maritime areas put collaboration over organisational and national borders to the test [31]. Rescue missions at sea are, however, performed by all available actors and not just those who are specialised in emergency response. Long distances and severe weather conditions have forced different parties to collaborate, regardless of whether they are publicly financed rescue services, voluntary financed services (e.g., Red Cross and Society for Sea Rescue), fishing fleets or the merchant navy [32].

Several catastrophes have led to updated and sharpened legislations in the maritime context. This was intensified after a series of shipping accidents with grave environmental damages during late 1960s and early 1970s. The subsequent investigations concluded that the main cause of these accidents was human errors, resulting from poor training and lack of inter-organisational competencies [33]. Subsequently, maritime mass-casualty disasters like the Scandinavian Star disaster in 1990 [34], the Estonia disaster in 1994 [35] and the Costa Concordia wreckage in 2012 [36] shed light on the need to put more efforts in inter-organisational education and training. Hence, the maritime domain started harmonising and regulating its educational standards worldwide.

The International Maritime Organisation (IMO)—a technical agency of the United Nations—was introduced in 1959 to harmonise standards of international maritime activities, including training and qualification of mariners who navigate ships. In 1978, the IMO launched its first educational standards—the Standards of Training, Certification and Watch Keeping for Seafarers (STCW). It was considered a breakthrough, as there were practically no international standards in maritime exercises [37]. However, the STCW did not succeed in harmonizing the already established national standards around the world. Several revisions seeking to repair the shortcomings of its predecessors were made, and new conventions were prepared including the recent Manilla amendments, STCW 2010. STCW 2010 highlighted and homogenised concepts of intra-organisational exercises for mariners. The STCW standardised security exercises in different scenarios such as attacks by pirates and operations in polar waters [38].

With regard to collaboration during emergencies in land-based contexts, there are no such standards as those provided by IMO. Some standardisation in crisis organisations and crisis work can be seen within certain covenants like the European Union. Nonetheless, there is still a lack of international as well as national standards in inter-organisational exercises. At the country level, there is some fragmentation. It has been reported about intra-organisational autonomy during emergency response operations from Sweden and Norway. The official evaluation following the 2011 terrorist attacks in Norway found, among other conclusions, that the emergency response had been insufficient, and that inter-organisational collaboration efforts in particular had been inadequate [10]. An article titled "Why is collaboration minimized at the accident scene?" reported on the difference between rhetoric and practice in connection with accident work. Collaboration is seen as a rhetorical ideal rather than something that is carried out in real life accident work. The results from the study showed that police force, fire fighters and prehospital healthcare staff have an intention to develop excellent forms of collaboration at the accident scene but avoid this because of asymmetries in standards and lack of incentives [39]. A study from a tunnel exercise in Norway showed a risk for competition within organisations with unclear hierarchies. A lack of clarity in the distribution of roles and power struggles was identified as potential risks affecting the handling of emergencies [40]. Short distances and high accessibility can also be reasons why land-based emergency staff can uphold a degree of autonomy. In contrast to operations at sea, land-based operations can most often be reached by all the required resources within reasonable time and normally do not need to rely on each other's willingness to cross professional borders [41]. Information exchange during land-based emergency response operations is also known to uphold a certain degree of autonomy. Each organisation in the response network has operational field units at different levels, different functional command structures, separate back offices and intraorganisational radio communications channels [42,43].

Scandinavian countries like Norway and Sweden have a lot of similarities and few differences. Because of a long border between Norway and Sweden, partly in populated areas, they have agreements on sharing resources and coordinating actions during emergencies. The response organisations of the two countries perform common inter-organisational exercises; moreover, the radiocommunication systems, "Nødnett" in Norway and "Rakel" in Sweden, have been connected since 2016 (MSB 2018). The open borders between the countries have a long tradition, based on similar languages, constitutional monarchies, democracy and governments appointed by a parliament in both countries. They share national emergency preparedness principles of responsibility, equality and proximity in their respective legislations [8].

The dependence between organisations at incident locations is well known, which justifies the need for inter-organisational exercises to be conducted regularly [39,44]. Obviously, there are similarities and differences between OFFSOs and ONSOs' collaborative strategies during emergencies. After decades of naval disasters, OFFSOs have developed harmonised standards, while, in the ONSO context, the routines are still fragmented. The ONSO context seems to be more affected by the perception of collaboration as a rhetorical ideal while OFFSOs are characterised by a willingness to cross organisational boarders. We still do not know if collaboration exercises differ in their impact on OFFSOs and ONSOs. Despite a lack of studies [45], inter-organisational exercises arranged by off-shore organisations (OFFSOs) as well as on-shore organisations (ONSOs) are practised at considerable cost, with the supposition that they contribute to learning and utility in real life disasters [3]. We aim to find similarities and differences in the outcome from inter-organisational exercises, in terms of learning and utility, in the maritime versus land-based contexts. The study is supposed to reveal context-specific challenges, weaknesses and differences in traditions of emergency preparedness in order to suggest how to improve the outcome from inter-organisational exercises [46].

## 3. Methods

Five studies from a Norwegian/Swedish research team was chosen to compare inter-organisational exercises arranged by ONSOs and OFFSOs. One of the studies provided data from three ONSO exercises. Four studies contained data from six OFSO exercises, four of the OFSO exercises were pooled into two datasets. The questionnaire used to collect data during all included exercises was the collaboration, learning and utility instrument (CLU) measuring collaboration, perceived learning and utility from 1 (strongly disagree) to 5 (strongly agree) on a Likert scale [1–5] (see the Supplementary Materials). The CLU scale has been applied in similar studies of crisis exercises [1,4,47,48]. Primarily, it was developed by a team of academic instrument-developers together with emergency practitioners from response organisations. The development was made in different steps based on Stein's [21] learning theories, which have their out spring from Klabber's [49] perspectives on how institutions learn, Meyer and Rowan's 1977 decoupling theory [50], Berlin and Carlström's theories on sequential, parallel and synchronous collaboration [51]. The collaboration dimension encompassed questions about the collaborative characteristics of an exercise. The learning dimension elaborated lessons learnt from collaboration during the exercises. The utility dimension determined if the exercise was perceived to be useful during real emergencies. In addition, questions were elaborated about experience and affiliation.

The CLU surveys were all distributed and collected from emergency personnel in connection with inter-organisational exercises at the included ONSOs exercises. Regarding the OFFSOs exercises, the survey was e-mailed to the participants from an e-mail list. The homogeneity of the 17 items showed a Cronbach's $\alpha$ of between 0.68 and 0.88. Statistical significance was established at $p = 0.05$, and all tests were two-tailed [52]. The analysis stems primarily from descriptive data and regressions (bivariate and multiple). Data were imported and analysed in Statistical Packages for the Social Sciences (SPSS) version 24.0.

### 3.1. The Context of the Survey

The studied inter-organisational exercises were full-scale field exercises. All the organisations that normally participated in the scenario, e.g., coast guard, sea rescue, police department, fire department and ambulance services, were engaged in each exercise. Moreover, the exercises aimed to improve inter-organisational collaboration, crossing organisational boundaries during accidents and disasters. The exercises took place in different parts of Norway and Sweden (Table 1).

**Table 1.** Distribution of respondents in the on-shore organisation (ONSO) and off-shore organisation (OFFSO) exercises.

| Published | Exercise/Scenario | Exercise Arrangement | Number of Organisations Involved in the Exercise | Number of Participants in the Study | Declined to Participate in the Study |
|---|---|---|---|---|---|
| [1] | Car ferry accident handled by ONSOS | ONSO | 3 | 39 | 2 |
| [1] | Fire at school | ONSO | 4 | 28 | 1 |
| [1] | Fire at work | ONSO | 4 | 27 | 3 |
| SUBTOTAL ONSOS: | | | 11 | 94 | 5 |
| [48] | 1. Maritime oil spill 2. Maritime search and rescue | OFFSO | 21 | 79 | 336 |
| [4] | 1. Fire at passenger ferry 2. Maritime search and rescue | OFFSOS | 22 | 53 | 11 |
| [50] | Maritime search and rescue | OFFSO | 8 | 30 | 32 |
| [49] | Maritime search and rescue | OFFSO | 27 | 90 | 472 |
| SUBTOTAL OFFSOS: | | | 78 | 252 | 852 |
| TOTAL ALL: | | | 89 | 346 | 857 |

### 3.2. Procedures

The survey included staff in different positions, for example, operational staff in the field, and staff officers from levels of management. Informed consent was obtained from the organisers, and each respondent was provided with written information. They were informed about confidentiality and the opportunity to withdraw from participation in the survey at any time. The data were collected with informed verbal consent.

Self-administered questionnaires collecting data were distributed. The questionnaires were coded for each exercise, and the completed questionnaires were collected anonymously by the authors. The questionnaires from the ONSOs were in paper form, and the later was outlined as web-based documents, while the questionnaires of the OFFSOs were web based.

## 4. Results

### 4.1. Participants

A total of 94 participants responded to the ONSO paper form survey and 252 to the web-based OFFSO survey.

The response rate was 95% for the ONSO survey and 49% for the OFFSO survey. The participants of the ONSOs and the OFFSOs had previously participated in 1 to 12 exercises (ONSOs M = 2.87, SD = 2.24; OFFSOs M = 3.53, SD = 1.64) before the study. Their age ranged from 25 to 49 years for the ONSOs and 18–55 for the OFFSOs. In the ONSO survey, all participants (100%) belonged to the public sector (police personnel, fire fighters and ambulance services). For the OFFSO survey, 86% belonged to the public sector, 10% to private sector and 4% to volunteer sector.

### 4.2. Collaboration

A majority of the ONSOs (75.6%, M = 4.07, SD = 0.72) and even more of the OFFSOs (88.1%, M = 4.44, SD = 0.87) considered the exercises to be focused on collaboration. Less than half of the ONSOs (41.4%, M = 3.13, SD = 1.11) and 63.5% of the OFFSOs (M = 4.00, SD = 1.19) experienced that the collaboration began without an unnecessary waiting time. Additionally, 42.6% (M = 3.34, SD = 0.98) of the ONSOs and 57.2% (M = 3.97, SD = 1.22) of the OFFSOs considered that the exercises encompassed alternative strategies to collaborate. Moreover, 53.2% (M = 3.70, SD = 1.01) of the ONSOs and 79.7% (M = 4.40, SD = 0.99) of the OFFSOs considered that staff who needed to practice collaboration were engaged in the exercises. Discussions took place after the practical activities in the studied exercises; however, 33% of the ONSOs thought these discussions were insufficient, and they wanted more seminar activities after the practical actions (M = 3.17, SD = 1.00). In contrast, 29.8% of the OFFSOs considered the discussions to be insufficient, while 20.6% remained neutral (M = 3.42, SD = 1.42). Out of all the ONSO respondents, 44.6% did not consider the exercises to be those they usually practiced (M = 2.82, SD = 1.30), while only 5.2% of the OFFSOs considered the same (M = 4.46, SD = 0.93). The mean for all items within the collaboration dimension was 3.52 (SD = 1.01) for the ONSOs and 4.06 (SD = 0.64) for the OFFSOs (Figure 1).

### 4.3. Learning

The majority of the ONSO and the OFFSO respondents replied that they learnt new things to a certain degree during the exercises (ONSOs M = 3.66, SD = 1.28; OFFSOs M = 4.16, SD = 1.16). Nearly half of the ONSOs (45.7%, M = 3.26, SD = 1.34) and more than half of the OFFSOs (66.6%, M = 3.84, SD = 1.14) considered themselves to have learnt new things about the organisations involved in the exercise. Less than a quarter of the ONSOs (21.4%, M = 2.72, SD = 1.03) and under half of the OFFSOs (42.5%, M = 3.22, SD = 1.30) learnt something about the concepts and acronyms used by the organisations involved as well as their communication patterns (ONSOs 22.4%, M = 2.74, SD = 1.05; OFFSOs 60.3%, M = 3.72, SD = 1.19). Quite a few of the ONSOs (17%, M = 2.79, SD = 0.88) and half of the OFFSOs (47.9%, M = 3.72, SD = 1.19) considered themselves to have learnt about prioritising activities. The mean for all items within the learning dimension was 3.03 (SD = 0.97) for the ONSOs and 3.67 (SD = 0.93) for the OFFSOs (Figure 1).

### 4.4. Utility

Most of the ONSO and the OFFSO respondents (84.1% and 77.7%) considered the exercises to be useful during actual emergency work (ONSOs M = 4.45, SD = 0.90; OFFSOs M = 4.20, SD = 1.11). Furthermore, they regarded the inter-organisational exercises as having an impact on their everyday work (ONSOs 61.7%, M = 3.69, SD = 1.16; OFFSOs 44.0% M = 3.26, SD = 1.22). It should be noted here that 27.4% of the OFFOS remained neutral. The exercises were considered to be more valuable for the command officers (ONSOs 58.5%, M = 3.54, SD = 1.30; OFFSOs 50.4%, M = 3.68, SD = 1.18) than for the operative staff in the field (ONSOs 30.8%, M = 2.97, SD = 1.22; OFFSOs 29.3%, M = 3.21, SD = 1.23). The mean for all items within the utility dimension was 3.66 (SD = 1.08) for the ONSOs and 3.59 (SD = 0.71) for the OFFSOs (Figure 1).

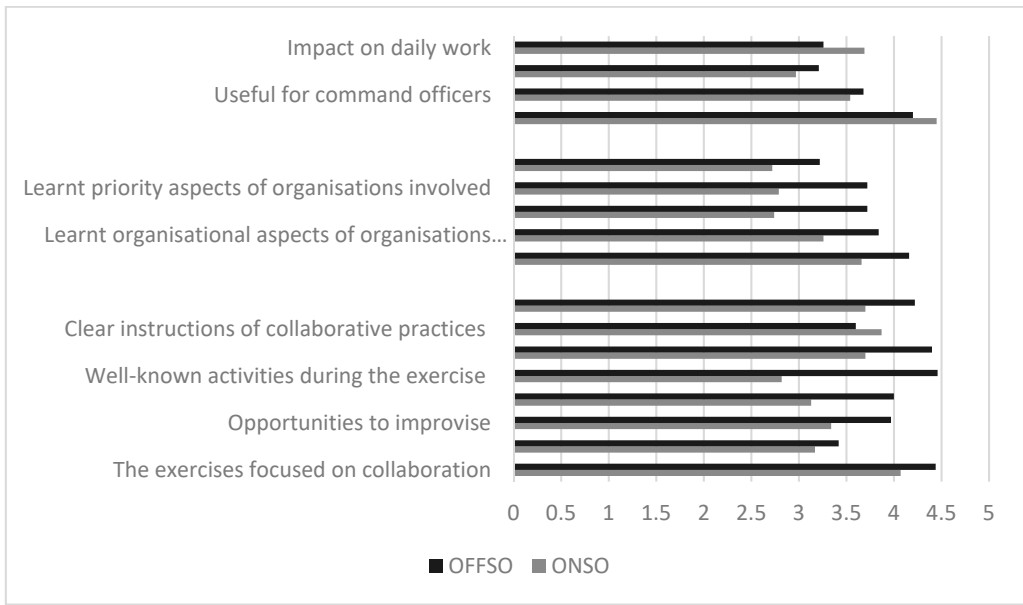

**Figure 1.** Mean values for the 94 ONSO and 252 OFFSO emergency personnel answering the collaboration, learning and utility instrument (CLU) scale, distributed in the dimensions utility (four items), learning (five items) and collaboration (eight items).

## 4.5. Bivariate Regressions

The causal effects of collaboration, learning and utility in the ONSO and OFFSO contexts were tested in a number of bivariate regressions. The collaborative dimension of the exercises was significantly correlated to the mean learning score across most of the items associated with the learning measurements in both exercise contexts. The strongest significant correlation on ONSOs was found between the item 'well-known activities during the exercise' and learning (R = 0.48), with this item explaining a significant proportion of variance in the mean learning score ($R^2$ = 0.23, F = 27.67, $p < 0.00$). The strongest significant correlation on OFFSOs was the item 'my point of view was regarded' and learning (R = 0.40), with this item explaining a significant proportion of variance in the mean learning score ($R^2$ = 0.16, F = 48.86, $p < 0.00$) (Table 2).

**Table 2.** Bivariate regression of items in the collaborative dimension of learning (sig. = $p < 0.05$).

| Bivariate Regression ONSO N = 94 OFFSO N = 252 | | | | | | |
|---|---|---|---|---|---|---|
| **Dependent Variables: Learning Independent Variables: Collaborative Characteristics of Exercises** | | | | | | |
| | | Pearson R | R-square | F-Value | T-Value | Sig. |
| 1. The exercises focused on collaboration | ONSO | 0.27 | 0.08 | 7.31 | 5.03 | 0.00 |
| | OFFSO | 0.33 | 0.10 | 30.92 | 5.56 | 0.00 |
| 2. Discussions immediately after the exercise | ONSO | 0.22 | 0.05 | 4.55 | 9.90 | 0.03 |
| | OFFSO | 0.16 | 0.02 | 7.08 | 2.66 | 0.01 |
| 3. Opportunities to improvise | ONSO | 0.21 | 0.04 | 4.18 | 9.19 | 0.04 |
| | OFFSO | 0.24 | 0.05 | 15.93 | 3.99 | 0.00 |
| 4. Collaboration was initiated immediately | ONSO | 0.39 | 0.16 | 16.81 | 10.19 | 0.00 |
| | OFFSO | 0.18 | 0.03 | 8.90 | 2.98 | 0.00 |
| 5. Well-known activities during the exercise | ONSO | 0.48 | 0.23 | 27.67 | 13.73 | 0.00 |
| | OFFSO | 0.11 | 0.01 | 3.46 | −1.85 | 0.06 |
| 6. Staff that needed to exercise participated | ONSO | 0.27 | 0.07 | 7.37 | 7.99 | 0.00 |
| | OFFSO | 0.14 | 0.01 | 5.11 | 2.26 | 0.02 |
| 7. Clear instructions of collaborative practices | ONSO | 0.43 | 0.19 | 21.33 | 8.69 | 0.00 |
| | OFFSO | 0.26 | 0.06 | 17.47 | 4.18 | 0.00 |
| 8. My point of view was regarded | ONSO | 0.42 | 0.18 | 19.85 | 6.02 | 0.00 |
| | OFFSO | 0.40 | 0.16 | 48.86 | 6.99 | 0.00 |

Some of the items in the learning dimension were significantly correlated to the usefulness items. The strongest significant correlation of ONSOs was found between the item 'learnt new things' and usefulness (R = 0.47, $R^2$ = 0.22, F = 26.35, $p < 0.00$). A somewhat weaker correlation was found for the OFFSOs for the same item (R = 0.35, $R^2$ = 0.11, F = 31.99, $p < 0.00$) (Table 3).

**Table 3.** Bivariate regression of items in the learning dimension of utility (sig. = $p < 0.05$).

| Bivariate Regression ONSO N = 94 OFFSO N = 252 | | | | | | |
|---|---|---|---|---|---|---|
| **Dependent Variable: Utility Independent Variables: Learning Characteristics of Exercises** | | | | | | |
| | | Pearson R | R-Square | F-Value | T-Value | Sig. |
| 1. Learnt new things | ONSO | 0.47 | 0.22 | 26.35 | 13.32 | 0.00 |
| | OFFSO | 0.35 | 0.11 | 31.99 | 5.65 | 0.00 |
| 2. Learnt organisational aspects of organisations involved | ONSO | 0.30 | 0.09 | 8.89 | 16.58 | 0.00 |
| | OFFSO | 0.32 | 0.10 | 27.44 | 5.23 | 0.00 |
| 3. Learnt communicational aspects of organisations involved | ONSO | 0.24 | 0.06 | 5.58 | 15.71 | 0.02 |
| | OFFSO | 0.29 | 0.08 | 21.82 | 4.67 | 0.00 |
| 4. Learnt priority aspects of organisations involved | ONSO | 0.15 | 0.02 | 2.09 | 13.48 | 0.15 |
| | OFFSO | 0.31 | 0.09 | 25.64 | 5.06 | 0.00 |
| 5. Learnt new concepts | ONSO | 0.14 | 0.02 | 1.89 | 16.04 | 0.17 |
| | OFFSO | 0.25 | 0.06 | 16.24 | 4.03 | 0.00 |

### 4.6. Multiple Regressions

Significant variables from the bivariate regressions were tested separately for the ONSOs and the OFFSOs in multiple regressions. In the case of ONSOs, the collaborative features together predicted 53% ($R^2$ = 0.53) of learning and for the OFFSOs, they predicted 25% ($R^2$ = 0.25) of the variance in learning. This meant that 47% and 75% of the predicted variance was non-unaccounted for in the case of ONSOs and OFFSOs, respectively. In the case of ONSOs, four variables were still significant: 'opportunities to improvise', 'well-known activities during the exercise', 'clear instructions for collaborative practices during the exercises' and 'my points of view were regarded'. The OFFSOs showed one still significant variable: 'my points of view were regarded'. The remaining variables displayed somewhat lower t-values and lacked significance on their own (Table 4).

**Table 4.** Significant variables of a multiple regression of items in the collaboration dimension of learning, (sig. = $p < 0.05$).

| Multiple Regression ONSO N = 94 OFFSO N = 252 | | | | | |
|---|---|---|---|---|---|
| **Dependent Variable: Learning Independent Variables: Collaborative Characteristics of Exercises** | | | | | |
| ONSO R = 0.73 R-SQUARE = 0.53 | | | | | |
| | Bivariate St.Beta. | Multi.regr.St.Beta. | Diff. | T-value | Sig. |
| 3. Opportunities to improvise | 0.21 | 0.04 | 4.18 | 9.19 | 0.04 |
| 5. Well-known activities during the exercise | 0.48 | 0.23 | 27.67 | 13.73 | 0.00 |
| 7. Clear instructions for collaborative practices. | 0.43 | 0.19 | 21.33 | 8.69 | 0.00 |
| 8. My point of view was regarded | 0.42 | 0.18 | 19.85 | 6.02 | 0.00 |
| OFFSO R = 0.52 R-SQUARE = 0.25 | | | | | |
| 8. My point of view was regarded | 0.40 | 0.29 | 0.11 | 4.78 | 0.00 |

In the next multiple regression, it was found that the items of learning predicted 26% ($R^2$ = 0.26) of the variance in usefulness in the ONSO and the OFFSO context. This meant that 74% of the predicted variance was still missing in the regressions. One variable was still significant, 'learnt new things' in the ONSO and the OFFSO contexts and 'learnt new concepts' in the OFFSO context. The other variables displayed moderate t-values and lacked significance on their own (Table 5).

**Table 5.** Significant variables of a multiple regression of items in the learning dimension of usefulness (sig. = $p < 0.05$).

| Multiple Regression ONSO N = 94 OFFSO N = 252 | | | | | |
|---|---|---|---|---|---|
| **Dependent Variable: Utility. Independent Variable: Learning Characteristics of Exercises** | | | | | |
| ONSO R = 0.50 R-SQUARE = 0.26 | Bivariate St.Beta. | Multi.regr.St.Beta. | Diff. | T-value | Sig. |
| 1.Learnt new things | 0.46 | 0.46 | 0.00 | 4.21 | 0.00 |
| OFFSO R = 0.52 R-SQUARE = 0.26 | | | | | |
| 1. Learnt new things | 0.47 | 0.32 | 0.15 | 4.71 | 0.00 |
| 5. Learnt new concepts | 0.63 | 0.13 | 0.50 | 2.04 | 0.04 |

## 5. Discussion

The OFFSOs considered the exercises to be focused on collaboration to a higher degree (88.1%) than the ONSOs (75.6%). In all items of collaboration, the OFFSOs scored a stronger result except on the item, 'if the actions were those they usually practiced'. Only 5.2% of the OFFSOs did not consider the actions to be those they usually practiced (ONSOs, 44.6%). A similar relation was found within the learning dimension. The majority replied they learnt new things, but the mean for all items within the learning dimension was higher for the OFFSOs (3.67) than for the ONSOs (3.03). The OFFSOs reported they learnt about communication patterns such as concepts and acronyms (60.3%) as well as prioritising (47.9%) to a higher degree than the ONSOs (22.4% and 17%). However, both OFFSOs and ONSOs showed equal results in the utility dimension. A few more participants in the ONSOs dimension perceived the exercises to be more useful during real life events than the OFFSOs (84.1% and 77.1%, respectively).

The multivariate regressions proved that collaboration had a higher degree of explanatory impact on learning for the ONSOs ($R^2 = 0.53$) than the OFFSOs ($R^2 = 0.25$). The impact of learning on the utility dimension was similar at sea and on dry land ($R^2 = 0.26$). The still significant items contributing to learning was 'my point of view was regarded' (OFFSOs and ONSOs). Additionally, 'well-known activities', 'clear instructions' and 'opportunity to improvise' had impact on learning in the ONSOs context.

The results of the study boil down to the finding that communication, learning new things and different points of view are regarded as important to achieve learning and utility from inter-organisational exercises. This calls for communicative and interactive processes during exercises [43]. There are, however, obstacles to achieving such a process. ONSOs repeat well-known activities and are known to be organisationally differentiated. OFFSOs seem to be more integrated than ONSOs, but they are restricted by international regulations. Furthermore, inter-organisational exercises often seem to strictly follow a predetermined manuscript, not allowing much room for improvisation. Consequently, exercises tend to restrict the learning and to be repetitive, drill-like and intra-organisational because of the lack of timeouts, spontaneous assessments and inter-organisational discussions, allowing for reflections on alternative ways to handle tricky situations [24,25,53]. The bivariate regressions, however, display that well-known activities had some impact on learning, especially in the ONSOs context ($R^2 = 0.23$ and OFFSO, $R^2 = 0.03$). A potential explanation can be the use of instrumental learning objectives, leaving little opportunity for improvisation [54]. This means that it is important to exercise well-known and repetitive everyday scenarios, and that new strategies may be developed to improve emergency responses. The result suggests that none of these extremes, (the unknown versus the familiar) can be excluded from inter-organisational exercises [42].

Lessons can be learned from inter-organisational exercises, both on- and offshore if they make room for the participants to talk informally before and after the training. In a recent study, the labelled side effect of the collaborative exercise was highlighted as more important than intra-organisational

practices [40]. The opportunity to get to know each other and discuss structures improved collaboration in real life emergencies. This means that even though the inter-organisational exercise per se seems to have poor results, it might lead to the creation of a collaborative culture among the participants. The ability to form and develop relationships might be more valuable than the planned formal learning.

The reason for the surprisingly high degree of learning from OFFSOs exercises, in terms of communication patterns, may be traced to the fact that crisis management in the maritime context is a high-tech sector. Due to several digital and analogous communication and positioning systems as well as international and national regulations, the complexity in training for an accident, or a disaster at sea will be very educational, even if the techniques and scenarios have been practised before. Moreover, if the size of the exercise is too big, it can hamper the possibility for learning, as different actors may focus more on their individual goals and solve discipline-specific tasks rather than on cross-border collaboration and the sharing of resources [40].

Another reason for the stronger learning outcomes in the maritime context may be different cultures at sea and on dry land. ONSOs practise manoeuvres unique to their intra-organisational context, e.g., weapons, advanced medical equipment, fire and rescue equipment from a strict professional and legal perspective. The OFFSOs qualifications, on the other hand, may primarily be dominated by seamanship together with professional practice. The seamanship is a common necessity for all collaborating OFFSOs, whereas the different professions, to some extent, are limited within the physical boundaries of the ship itself. A second factor that is important for the evolvement of cultural bonds is a tradition of spontaneous communication between crews at sea. On dry land, we have bystanders, i.e., the public, and first responders, but in the maritime context all parties are expected to act as first responders, regardless of whether they are publicly financed rescue services. In densely populated areas on land, it is easier to hide behind the crowd during an emergency than it is at sea. Petrenj studied collaboration during accidents on land and highlighted that the main obstacles to collaboration was a lack of incentives and ambiguity in roles and expectations [55]. In the maritime context, actors have proven to be aware of a common and core value of understanding and agreement [56]. Such a value can offer synergies focused on the big picture rather than just prioritising specific tasks they are trained for [19]. In contrast to ONSOs, collaboration at sea seems to be part of the preparedness for everyday work and the ethos of seamanship.

### 5.1. Practical Implications

This study shows collaboration and learning may be obtained from exercises based on already well-known activities. In this study, well-known activities were combined with opportunities as independent factors for learning. These results can inspire exercise planners to combine the unknown with familiar scenarios. The results may also encourage planners to use timeouts, seminars and repetitions of actions in order to elaborate highly efficient types of inter-organisational collaboration. The use of scripts may hamper learning and utility. This can be addressed by increased the focus on improvisation, together with a new training methodology. Crossing organisational borders during exercises can contribute to a culture that stimulates collaboration, learning and utility in actual emergency work.

### 5.2. Limitations

There are limited studies on exercises and even fewer conducting meta-analysis of exercises. This study is based on five studies on nine exercises and the ONSOs were collected from only one study. In order to verify the results, several more context specific explorations have to be performed. Furthermore, studies on inter-organisational exercises on land and in the maritime context should be extended to other areas such as the aerospace industry and other security-intensive contexts. Even though Norway and Sweden, where the data for this study were collected, have a lot of similarities, exercises from different cultural contexts can provide important results, contributing to the understanding of how to improve learning and utility from inter-organisational exercises.

The poor participation for the OFFSO survey may be due to a lack of research interest among participants, but also the lack of influence of big scale exercises. It can also be related to intra-organisational focus- and sector-specific objectives. Another reason may be the web-based questionnaires. Compared to paper questionnaires, web-based questionnaires have proven to be less responsive in the Scandinavian context [57].

**Supplementary Materials:** The following are available online at http://www.mdpi.com/2071-1050/12/14/5604/s1, Figure S1: CLU-instrument.

**Author Contributions:** Conceptualization, all; methodology, E.C. and J.L.S.; writing—original draft, E.C.; writing—review and editing, all; supervision, E.C. All authors have read and agreed to the published version of the manuscript.

**Funding:** This research received no external funding.

**Conflicts of Interest:** The authors declare no conflict of interest.

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
