# Peer review of "Inter-Organisational Exercises in Dry and Wet Context—Why Do Maritime Response Organisations Gain More Knowledge from Exercises at Sea Than Those on Shore?"

_sustainability, doi:10.3390/su12145604_

Round 1

Reviewer 1 Report

Very good paper written using academic standards.

Paper is very informative and well structured, it has all necesarry elements and the study is interesting.

The only minor suggestion - please link the study contents with Sustainability publishing area.

Author Response

Good point, the content of the study is, in the new version linked to the Sustainability publishing area. Please see new intro.

Reviewer 2 Report

This is an interesting and original paper. The findings are well supported and conclusions well drawn. The abstract and keywords present well what to expect. However, the quality of the presentation can be improved. The introduction provides an extensive overview of references, including high level journals from a number of fields, including humanities, but lacks in summarising at the end how the gaps are addressed in the current paper. This should be improved. Also, the results should be better visualised: less CAPS in the tables and the paragraphs 4.1-4.3 show a lot of numbers in plain text which would be better presented in at least a table but better a diagramm.

Author Response

Good point, the background section in the new version has been extended in order to address the gaps.

Thanks for the comment about the tables, they are updated and a figure (diagram) is added to the new version.

Reviewer 3 Report

Inter-organisational exercises seems to be important and interesting topic. So in my opinion, after minor revisions in line with the following four reviewer's comment, the manuscript shall be accepted.

  1. Before the authors claim that "Inter-organisational exercises are supposed to help authorities to become better at handling accidents, crises and disasters", authors should show how better handling can contribute sustainable issues, which relates this journal's scope.
  2. And authors must state their research purpose clearly in the Introduction section. Based on the present manuscript, to find similarities and differences? to map similarities and differences? For the reviewer, the conclusion doesn't seem to "map" similarities and differences enough.
  3. "CLU" must be spelled fully when the word first appears. And authors must write how they chose or create 17 questions in three dimensions more detail. Were those based on the existing papers or reports? Or how can authors claim their validity in this context?
  4. In the Discussion, some sentences are seems to be same with ones in the Results section. So, authors must try to generalize their results in the Discussion section as much as possible.

Author Response

1.Good point,in the new version the content of the study is linked to the Sustainability journals scope, please see new intro.

2.Map is changed to find in the purpose.

3.Changed in the new version, CLU is spelled fully first time the abbreviation appears and more info about the questionnaire and its origin is added.

4.Based on a comparison of the background and the discussion, some of the text in the discussion section is updated in the new version, thanks for the comment.

Round 2

Reviewer 2 Report

The paper improved according to the previous version: a little in terms of references and well in term of the look of the tables, which were my revision points.